# Experimental and Bioinformatic Approaches to Studying DNA Methylation in Cancer

**DOI:** 10.3390/cancers14020349

**Published:** 2022-01-11

**Authors:** Angelika Merkel, Manel Esteller

**Affiliations:** 1Bioinformatics Unit, Josep Carreras Leukemia Research Institute (IJC), 08916 Barcelona, Spain; 2Cancer Epigenetics Group, Josep Carreras Leukemia Research Institute (IJC), 08916 Barcelona, Spain; 3Centro de Investigación Biomédica en Red Cáncer (CIBERONC), 28029 Madrid, Spain; 4Institucio Catalana de Recerca Avançats (ICREA), 08010 Barcelona, Spain; 5Physiological Sciences Department, School of Medicine and Health Sciences, University of Catalonia, 08017 Barcelona, Spain

**Keywords:** DNA methylation, cancer, methods, software, computational analysis

## Abstract

**Simple Summary:**

Aberrations of normal DNA methylation patterns are observed in many cancers and are associated with chromatin alterations, changes in gene expression and genomic instability, making the study of DNA methylation paramount to our understanding of cancer biology and evolution and the development of biomarkers. Here, we present an overview of genome-wide approaches for the analysis of DNA methylation with relevance to cancer research and clinics.

**Abstract:**

DNA methylation is an essential epigenetic mark. Alterations of normal DNA methylation are a defining feature of cancer. Here, we review experimental and bioinformatic approaches to showcase the breadth and depth of information that this epigenetic mark provides for cancer research. First, we describe classical approaches for interrogating bulk DNA from cell populations as well as more recently developed approaches for single cells and multi-Omics. Second, we focus on the computational analysis from primary data processing to the identification of unique methylation signatures. Additionally, we discuss challenges such as sparse data and cellular heterogeneity.

## 1. Introduction 

DNA methylation, as supposed to other epigenetic marks, represents a direct modification of the genome, that is, the addition of a methyl group at the 5th Carbon of the cytosine base. In mammalian genomes predominantly found at CpG dinucleotides, it is an essential mark for normal mammalian development and a defining feature of cellular identity [1]. Aberrant DNA methylation patterns have been observed in numerous diseases, particularly in cancer where global hypomethylation and promoter hyper-methylation are characteristic of the disease [2]. The combined potential to identify subtypes in genetically highly heterogenous cancers and its relative stability during cell proliferation has made DNA methylation an attractive mark for cancer research and diagnostics [3].

DNA methylation is thought to alter chromatin structure in concert with other epigenetic marks, such as histone modifications, transcription factors, etc., and henceforth modify transcriptional potential or, in other words, regulate gene expression. However, the associated biological phenomena are manifold. Initially, much attention was given to the excessive methylation of CpG-rich promotors that occurs at many cancer-related genes. Hyper-methylation in these regions is associated with repression of transcription, whereas in normal tissues, these are generally unmethylated and associated with active gene expression [4,5]. More recently, technological advances of genome-wide high-throughput technologies have revealed that a large proportion of regulatory elements for which DNA methylation marks tissue specificity are located in CpG-poor regions far away from genes [6]. Additionally, DNA methylation may be altered dynamically, affecting TF-binding [7] and thus gene expression in an environmentally dependent manner. Finally, DNA methylation within large hypo- or partially methylated regions occurs in a stochastic manner and follows heterochromatic domains—a phenomena that ultimately has been observed not only in cancers but also in normal cells and that has been associated with cell proliferation history [8,9].

Not only have recent technological advances allowed analysing the smallest amounts of input material down to individual cells but also made it possible to simultaneously capture information from multiple omics essays. These, together with the existing technologies for high-throughput genome-wide analysis, have resulted in a large diversity of computational tools to process and analyse methylation data. Here, we provide an overview of approaches for DNA methylation analysis. We do not attempt to include all available tools but rather have selected the most popular methods to showcase the breadth and depth of information that this epigenetic mark provides for cancer research. We first review established and recent experimental approaches used in the field; second, we focus on bioinformatic data processing and analysis. Finally, we discuss outstanding challenges and future perspectives.

## 2. DNA Methylation Assays

In short, DNA methylation can be interrogated based on three approaches: (1) bisulfite conversion-based (chemical conversion of unmethylated cytosine to uracil, whereas methylated cytosines are not converted); (2) methylation-sensitive-enzyme-restriction-based (MSRE, restriction site includes methylated or unmethylated cytosine); or (3) affinity enrichment-based (active binding site includes methylated cytosine). Once established, the methylation signal is then analysed by either DNA hybridization or sequencing. To date, analyses of bisulfite-converted DNA by microarray or Next Generation Sequencing are the most commonly employed approaches for methylome studies (Figure 1A). MSRE approaches are limited to the existing restriction sites; therefore, their resolution is limited to fragment size, which particularly for CpG-poor regions, is low. Affinity-based approaches, such as methylated DNA immunoprecipitation sequencing (MeDIP-seq), methylated DNA capture by affinity purification (MethylCap-seq) or methylated DNA binding domain sequencing (MBD-seq), enrich for CpG-rich regions. Although they avoid the DNA damaging bisulfite-treatment, they are more labour intensive, requiring an additional step to retrieve the DNA fragments bound by the respective proteins, and the interpretation of the DNA methylation signal is somewhat difficult.

Whole Genome Bisulfite Sequencing (WGBS) constitutes the gold standard for DNA methylation analysis, providing single-base-pair, genome-wide resolution at a coverage of up to 95% of all CpGs in the human genome (~28 × 10^6^ CpG). More cost efficient due to less sequencing required, Reduced Representation Bisulfite Sequencing (RRBS) enriches for CpG-rich fragments via *Msp1* restriction and fragment size selection, covering mainly CpG islands and providing coverage of 2–4.5 × 10^6^ CpGs. By far, the most popular approach, due to its low cost and fast turnaround, is the analysis of a fixed set of CpG probes via DNA hybridization microarrays. The widely distributed Illumina BeadChip microarray can cover 27,578 (27K), ~450,000 (450K) [10], or in its latest generation, ~850,000 human CpGs (EPIC array) [11]. Probes mostly include CpG-rich regions, gene promoters, and known *cis*- regulatory elements (EPIC array). Importantly, for its potential clinical use, the described DNA methylation microarrays perform well for DNA extracted from paraffin-embedded (FFPE) sections [12]. Costs for studies with high sample through-put can be reduced by using targeted sequencing approaches or custom design microarrays (for comprehensive reviews see [13,14]. With the increased application of single-cell and single-cell multi-omics approaches, studies using approaches based on bulk DNA sequencing (WGBS, RBBS) have seemingly declined in popularity compared to microarray-based studies. (Figure 1B–D) However, bioinformatics tools for analysing bisulfite sequencing data, such as Bismark [15], have retained traction since processing of single-cell DNA methylation data is mostly the same as for bulk DNA (Figure 1B).

Bisulfite-based approaches do not distinguish between 5-methyl-(5mC) and 5-hydroxymethyl Cytosine (hmC) modifications. 5-hmC is generated by oxidation of 5mC by TEN-Eleven Translocation (TET) enzymes and was initially perceived as intermediate in a replication-independent demethylation pathway leading to unmodified cytosine [16]. Studies, however, have reported abundant tissue specific stable hydroxymethylation in neurons and embryonic stem cells making 5-hmC, an epigenetic mark in its own right [17,18]. Additionally, loss of 5-hmC has been observed hematopoietic malignancies and solid cancers [19,20]. Hydroxymethylation can be assessed via oxidative bisulfite sequencing (oxBS) or TET-assisted bisulfite sequencing (TAB-seq). oxBS involves the specific oxidation of 5hmC to 5-formylcytosine (5fC) and conversion of the newly formed 5fC to uracil (under bisulfite conditions) [21]. TAB-seq involves β-glucosyltransferase (β-GT)-mediated protection of 5-hmC (glucosylation) and recombinant mouse Tet1(mTet1)-mediated oxidation of 5-methylcytosine (5-mC) to 5-carboxylcytosine (5-caC). After the subsequent bisulfite treatment and PCR amplification, both cytosine and 5-caC (derived from 5-mC) are converted to thymine (T), whereas 5-hmC reads as C [22].

An alternative approach to detect DNA modifications (5mC, hmC, 6mA and others) without involving aggressive chemical treatment is direct long-read sequencing via nanopore sequencing technology. Long reads also allow to study the co-occurrence of base modifications along individual molecules, as well as their phasing with genetic variants, opening up opportunities in exploring epigenetic heterogeneity (see Section 4.2.). Nanopore sequencers (MinION, GridION and PromethION) measure ionic current fluctuation of single-stranded nucleic acid polymers when passing through a biological nanopore [23]. Each nucleotide, including their chemical modifications, exhibits different alterations of the current, and therefore the sequence of bases can be inferred from the specific patterns of current variation. Modifications are inferred as differential patterns from modified and unmodified base calls. Although nanopore basecalling has significantly improved in recent years and there have been several proof-of-concept studies (see [24] for a review), prediction of methylation states from basecalls are still somewhat suboptimal [25]. This has prompted other innovative approaches such as enzymatic methyl-seq (EMseq), which employs C-T conversion via 5-carboxylcytosine combined with nanopore sequencing to increase prediction accuracy and show good accordance with standard approaches such as WGBS [26].

### 2.1. Single-Cell and Single-Cell Multi-Omics Approaches

During the previous two decades, NGS and microchip technology have elevated DNA methylation studies to yield higher and wider genomic resolution and throughputs of hundreds of samples. More recently, however, technological development has focused on lower DNA input, such as single cells, and the simultaneous incorporation of other “Omics” assays, so-called “multi-Omics methods” (Figure 1C,D).

The first single-cell protocol, scRRBS [27], was established as an adaptation of bulk DNA sequencing for low input material but had only limited genomic coverage (40% of conventional RRBS) and showed excessive PCR duplicates (a fall-back which was later tackled by Q-RRBS with the addition of unique molecular identifiers (UMI) to each initial DNA fragment [28]). scBS-seq [29] implemented post-bisulfite adapter tagging (PBAT) and increased coverage to about 18% of genomic CpGs. Here, sequencing adapters are added to the DNA after bisulfite treatment to prevent loss of fragmented DNA, which is a result of the aggressive BS reaction. It was quickly followed by scWGBS [30] that also utilized PBAT, although without a pre-amplification step that was used by scBS-seq and allowed the preservation of strandedness and reduced amplification bias. Other approaches, such as single nucleus methylcytosine sequencing (snmC-seq), improved the recovery of bisulfite-converted, single-stranded DNA during library preparation [31], or such as sci-Met, added high-throughput single-cell processing by combinatorial indexing [32].

Single-cell multi-omics approaches that included methylation analysis were developed starting in 2016: scMT [33] and scM&T to interrogate DNA methylation and the transcriptome simultaneously [34]; and scTrio to interrogate methylome, transcriptome and copy number variation at the same time [35]. DNA and RNA are physically separated prior to bisulfite treatment and are analysed by scBS-seq or scRRBS, scWGS and Smartseq2, respectively. Shortly after, the Nucleosome Occupancy and Methylation sequencing protocol (NoMe-seq) which interrogates open chromatin, nucleosome positioning and DNA methylation was adapted for single-cell analysis with scNOMe-seq [36] and further combined with Smartseq2 for transcriptome analysis with scNMT [37] (Figure 1D). Other developments of multi-omics approaches are directed towards higher sample throughput and systems that prevent DNA loss, such as single tube reactions and multi-fluid systems (for an extensive review on single-cell/ single-cell-Omics methods, see [38]).

### 2.2. Cell-Free Circulating Tumour DNA (ct) from Liquid Biopsies

Analysis of cell-free circulating tumour DNA (ct) from liquid biopsies provides a minimal-invasive approach for the study and monitoring of tumour evolution. Epigenomic analyses, including DNA methylation, can significantly contribute to information gained from genomic analysis of cf-DNA and have outperformed classifications and cell-of-origin assignments based on SNP and CNV calling [39]. Ct-DNA is highly fragmented (mostly 130–160 bp fragment length) and may make up 3–90% of the total cell-free DNA, depending on cancer type and stage [40], which requires highly sensitive analysis methods such as deep sequencing (BS-seq) or targeted approaches (hybrid capture, PCR). Alternatively, pooled cf-DNA extractions have been used for microarray analysis to reach sufficient amounts of required inputs [41]. Additionally, affinity-based approaches, specifically cfMeDIP-seq, have proven particularly successful, as they only require minute amounts of input material [42]. Due to the peculiarities of the input material, studies generally first establish reference panels to identify significant marker loci, then subsequently assay these [39,40].

## 3. Processing of DNA Methylation Data

All data processing, whether based on bisulfite sequencing, bisulfite microarrays or affinity enrichment, starts with an initial step of raw data quality control (Table 1). From here, sequence-based approaches perform trimming of unwanted bases from the reads, such as sequencing adapters or unwanted bases resulting from enzymatic end repair. However, alignment for BS-seq reads needs to consider the bisulfite-induced conversion of un-methylated Cytosine to Uracil (and subsequently to Thymine through PCR amplification). As such, BS aligners either perform a wild-card alignment against C or T equally (e.g., BSMAP [43]) or, more common, align against a converted and un-converted version of the reference genome (so-called ‘three-letter aligners’ such as Bismark [15], BS-seeker [44], gem3 [45]; for a recent benchmark of BS aligners, see [46]). MBP-seq approaches, since they do not undergo bisulfite treatment, require only standard genomic read alignment tools. The post-alignment removal of PCR duplicates (usually by identifying reads with the same start and end coordinates) is performed for all approaches involving sequencing, but due to the enrichment step, approaches such as RRBS or MeDiP-seq and MethylCap only remove duplicates above a certain coverage threshold. Since naturally occurring single-nucleotide variants (SNPs) affect the methylation estimates from BS-treated DNA, known SNPs are mostly filtered out. Some processing pipelines such as gemBS [45] incorporate SNP calling from BS-seq to detect additional SNPs.

Finally, methylation levels are estimated from read coverage. For BS-seq data, this is commonly calculated from reads containing unconverted (=methylated) or converted (=unmethylated) cytosines as the proportion of unconverted cytosines over all counts (unmethylated + methylated). More accurately, this can be estimated in a probabilistic manner, taking into account bs-conversion rates and sequencing errors as well as a beta-binomial distribution dependent on read counts, although in practise, the differences are marginal and become only significant at low read coverage [63]. For MeDIP-seq and MethylCAP, methylation levels are estimated as local enrichment of reads, which is derived from a normalized signal against the background [56,64].

Since they are based on hybridization technology, data from microarrays do not require any read alignments. For the Illumina BeadChip platforms, the initial chip image is internally processed inside the scanner. The output signal, the chips’ two colour channels (red and green), is then subsequently background corrected (quality controls probes) and normalised using a variety of methods [65]. The popular analysis package minfi includes several normalizations, amongst them, for example, ssNoob, which adjusts for technical variation across platforms [60]. Finally, methylation levels are inferred from the ratio of both colour channels as beta value or its log ratio, the M-value.

## 4. Analysis of DNA Methylation

### 4.1. Exploratory Data Analysis and Sparse Data

As a first step after the primary data processing, exploratory analysis usually involves visualizing similarities between samples to check for technical biases (batch effects) or phenotypes (Table 2). Principle component analysis (PCA) is useful for visualizing data spread along individual components of sample variance. In the case of complex data, such as single-cell data, dimensionality reduction methods such as classical multidimensional scaling (MDS), t-distributed stochastic neighbour embedding (t-SNE), or negative matrix factorization (NMF) have proven successful [31,32]. Further, clustering approaches such as k-means or hierarchical clustering (un-/supervised) are useful in identifying meaningful groupings such as cancer sub-types, treatment conditions or cell populations. Specialized algorithms have been used for single-cell data analysis (e.g., DBSCAN [32]) and multi-omics data [66,67]. Multi-omics clustering leverages data from multiple assays and allows for a more comprehensive insight into population structure, while at the same time facing computational and statistical challenges of multi-dimensional integration (for in-depth evaluation of different methods, see [66,67]).

Compared to bulk DNA, single-cell data are sparse and discrete. As there are theoretically only two copies of any given DNA fragment present and bisulfite treatment aggressively attacks those, any failure of capturing a particular fragment or sequencing error results in data loss. Sequencing data from single-cell DNA typically suffers generally low mappability and reduced heterogenic genomic coverage. As a result, methylomes from single-cell data are often composed of 10–100s of individual cells, or the methylation signal is summarized over genomic regions [30]. To achieve more even genomic coverage and facilitate downstream analysis, several algorithms have been developed that leverage the correlation of methylation levels across neighbouring CpG as well as information from across-cells information. For example, Melissa [75] and Epiclonal [76] use local regression models combined with a (Bayesian) model prior to predict latent methylation profiles of genomic regions. DeepCpG [77], on the other hand, employs deep learning (neural networks) to predict methylation levels based on sequence composition (Table 2).

### 4.2. Deconvolution of Cellular Heterogeneity and Estimating Tumour Purity

When investigating bulk DNA, a major convoluting factor in analysing differential methylation is that variance amongst conditions might be caused by factors not related to differences in cellular phenotypes but rather to differences in the cell type composition of samples. Surrogate variance analysis (SVA) is an established method to remove unwanted variation (batch correction) of unknown origin, and it can similarly be applied to correct for the difference in cell-type composition (Table 3). Other reference-free and semi-reference-free approaches employ methods such as NMF and recursive Quantile Projection (QP) which have been used to estimate cell type proportions. However, when the contributing cell types are known and/or reference data are available, reference-based approaches for cell-type deconvolution such as robust partial correlations (RPC), support vector regression (see CIBERSORT/METHYLCIBERSORT) or constrain projection (see Houseman CP) are preferrable [78,79]. Naturally, the success of the deconvolution highly depends on the quality and applicability of the reference and the knowledge about which cell types to expect. Low or inappropriate reference data can lead to biased results. Classically, pure sorted cell populations have been used as refences, but single-cell data are increasingly incorporated. Teschendorff et al. [80] even developed deconvolution of bulk WGBS using scRNA-seq data.

The latter class of algorithms has been applied predominantly to blood since it is a medium frequently used for research (and diagnostics) and is known for a cellular composition that readily changes depending on a variety of factors. In cancer research, however, an important issue is to estimate tumour-purity by accounting for non-tumoral cells, but often tumour reference data are rarely available. Here, the package HEpiDiSH uses reference data from immune cells, fibroblasts, epithelial cells and adipocytes to infer the tumour model (e.g., for oral and breast cancer) [84]. Other packages such as MethylResolveR [86] have concentrated on using a set of distinct immune cell types or, similar to METHYLCIBERSORT (based on CIBERSORT) [85], have additionally established a large cancer reference set from cell lines to demonstrate that the type and proportion of contributing/invading immune cells are associated with survival and other characteristics.

Any deconvolution of heterogeny as described above has is based on differentially methylated, cell specific informative sites, which have to be identified and extracted upfront from the global set (see Section 5.1). Alternative approaches utilize information from neighbouring CpGs that are co-located within sequencing reads from bisulfite sequencing data. As mentioned before, methylation patterns tend to spread across a region such that there is high correlation between neighbouring CpGs in normal cells. Cancers have aberrant methylation, showing higher variations in DNA methylations than normal cells. By stratifying sequencing reads from bisulfite sequencing data into concordant reads (all CpGs are either methylated or unmethylated) and discordant reads (CpG have disordered methylation patterns), the proportion of discordant reads can be used to estimate the tumour proportion in a given sample—an approach that has been successfully implemented in predicting the tumour-derived cell free-DNA fraction in human cancer plasma [87]. Similarly, measures such as epipolymorphism and methylation entropy are based on epiallele frequency (epiallele—unique combination of CpG methylation states within a read) and have also been used to quantify within sample heterogeneity [89].

## 5. DNA Methylation Signatures

### 5.1. Differential Methylation

Identifying meaningful specific methylation signatures is the ultimate goal of methylation analysis. Differential methylation can be described for single sites (DMC—differentially methylated CpGs; DMP—differentially methylated probes), sets of adjacent sites (DMR—differentially methylated regions), or pre-defined genomic regions such as tiling windows, promoters, enhancers, etc. (Table 4).

For microarray derived beta-values, significant differences between two groups of samples are commonly estimated based on t-statistics (t-test, Welsh-test, Permutation test) or moderated t-statistics (Empirical Bayes) since their distribution is approximately Gaussian. Methylation values derived from count data such as methylated/unmethylated sequencing reads follow a binomial distribution and are modelled by either a beta-binomial or negative-binomial distribution variance across samples estimated by a dispersion parameter (edgeR, DSS, MethylKit). For analysing multiple samples and/or to include covariates, regression analysis is the natural choice. Once significance is established (e.g. *p*-value < 0.01), relevant sites are usually selected by a minimum threshold of absolute differences between mean methylation values which may depend on the phenotype in question. For example, many smoking-associated DMCs show differences as low as 5%, whereas most cancer-associated DMCs exhibit differences much larger (25–30%) [78].

In practice, changes in methylation are typically estimated for differentially methylated regions (DMRs) rather than individual DMCs. This reduces data dimensionality and increases the power of detection by employing nearby CpGs. In the case of pre-defined regions, the methylation signal is simply summarized over the entire region, and statistical testing, similar as for single sites, is applied (e.g., MethylKit [88], RnBeads [92], edgeR [96], MethylSig [90]). Alternatively, de novo DMRs can be defined as an extension of DMCs, where a DMR constitutes a region containing a minimum number of DMCs at a maximum distance and minimum/maximum absolute length (e.g., DSS package [97]). Other tools take into account the correlation between nearby CpGs. Bumbphunter [98] (implemented in the minfi package) first fits a linear regression model for each locus and then smooths the coefficient within clusters along the genome to identity bumps, i.e., DMRs. Similarly, DMRcate [94] first establishes local moderate t statistics (limmas t^2^s) and then applies a Gausschian kernel for smoothing within a specific window—a method the authors claim to be platform agnostic and to remove bias derived from sparse and irregularly spaced CpGs. Dmrff [95] derives subregions from stretches of DMCs, evaluates and adjusts them, and then combines the most significant into candidate DMRs which are then evaluated again.

### 5.2. Methylome Segmentation and the DNA Methylation Landscape

Another approach to describe methylation signatures takes into account larger scale methylation features in the form of genomic segments that have been shown to be associated with certain chromatin states and 3D structures. Hidden Markov Models (HMMs) have been implemented in several applications to identify regions of similar methylation states (Table 5). For example, Stadler et al. (2011) [7] implemented a three-state HMM that identified unmethylated regions (UMRs), lowly methylated regions (LMRs) and fully methylated regions (HMRs), which corresponded to unmethylated CpG islands, short CpG-poor regions with intermediate methylation and the remaining bulk of the genome, respectively. UMRs were associated with open chromatin and active transcription start sites, whereas LMRs were identified as active enhancers. A two-state HMM by Song et al. (2013) [99] identified hypo- and hypermethylated regions recorded in MethBase, a public database of tissue-specific DNA methylation features. Longer hypo-methylated regions of several kilobase pair lengths were coined ‘DNA methylation valleys (DMVs) and canyons’ and have been implicated in developmental processes and cancers, such as leukaemia and advanced prostate cancer [100,101,102].

On a larger scale, at the size of tens of kilobase and megabase pairs, so called “hypomethylated domains” or “partially methylated domains” (PMDs) are associated with heterochromatin and transcriptional silencing and coincide with topologically associated domains (TADs) and lamina-associated domains (LADs) [9,104]. The stochastically occurring methylation in these domains and the associated global hypomethylation are characteristic for many cancers. However, several studies have shown they also occur in normal tissues, such as fibroblast, adipocytes or mature lymphocytes [6,8].

## 6. Downstream Analysis: Interpretation and Application of DNA Methylation Signatures for Research and Clinics

Interpretating DNA methylation signatures generally involves other types of omics data for additional downstream analysis. For example, the correlation of DNA methylation with gene expression via RNA-seq or expression arrays allows assessing the phenotypic impact of epigenetic modifications and insights into biological processes via pathway and/or network analysis. Associating DNA methylation states with histone modifications and TF motif binding (e.g., from ChIP-seq experiments) as well as chromatin accessibility (e.g., via DNAseq, ATAC-seq) and conformation (e.g., Hi-C seq) allows to unravel mechanisms of gene regulation (see https://epigenie.com/epigenetic-tools-and-databases/, last accessed on 1 January 2022, for a list of epigenetic tools and databases for downstream analysis and visualization).

A vital output of DNA methylation analysis for cancer diagnostics is the identification of novel and clinically relevant subtypes. Recent statistical approaches for integrative multi-omics analysis (including similarity- and correlation-based, Bayesian, fusion and other multi-variate methods) have greatly improved subtyping of cancers and feature selection identifying novel biomarkers and driver genes [105]. Machine learning algorithms, such as random-forest or neural networks, have enabled the classification of brain tumours [106] and sarcomas [107] and even the assignment of primary tumour sites for metastases of unknown origin [108]. Similarly, important for the clinic, they have been applied to predict responses to pharmacological [109] and cellular immunotherapy [110] and model patient survival [111].

## 7. Conclusion and Remaining Challenges

Next generation sequencing and microarray technology have allowed to interrogate DNA methylation at unpreceded genomic resolution and sample throughput, while recent low input and single-cell technologies have enabled interrogation of cell-free DNA and rare-cell populations. Through computational tools, these data have yielded insights into the interplay of DNA methylation and chromatin structure and have greatly improved our understanding of cancer biology and evolution. Together, they have greatly aided the molecular characterisation, classification and ultimate detection of cancers and their tissue of origin. While genome-wide methylation analyses are paramount for the development of biomarkers, for clinical practice, they are further informative to determine treatment resistance and predict cancer risk, fragility, and mortality rates.

Challenges remain in areas of deconvoluting cellular heterogeneity, where clonal or and/or cellular heterogeneity is excessive, or reference data are sparse or not available. Furthermore, cell–cell interactions represent an important characteristic in tumour biology and remain largely unaccounted for by most algorithms.

## Figures and Tables

**Figure 1 cancers-14-00349-f001:**
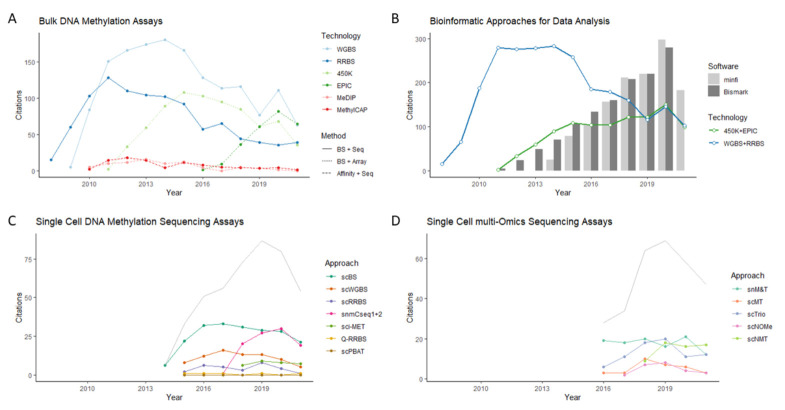
Popularity (Pubmed Citations) of DNA methylation approaches. (**A**) Bulk DNA methylation assays. (**B**) Experimental and bioinformatic approaches for bisulfite sequencing (WGBS + RRBS) and bisulfite arrays (450K and EPIC). Bismark, popular BS-seq analysis tool; Minfi, popular microarray analysis tool. (**C**) Single-cell DNA methylation sequencing assays. (**D**) Single-cell multi-omics sequencing assays.

**Table 1 cancers-14-00349-t001:** Data analysis and methylation calling.

Description	Software	Bulk BS-Seq	scBS-Seq	AE-Seq	BS- Arrays	Ref
Quality control	FastQC	yes	yes	yes		[47]
Adapter/end-base trimming	TrimGalore	yes	yes			[48]
BS-aware read alignment	BISMARK, BS Seeker2, gemBS, BSMAP	yes	yes			[15,43,44,45]
Remove PCR duplicates	PicardTools	yes	yes	yes		[49]
Variant calling	gemBS, Bis-SNP, GATK	yes				[45,50,51]
Methylation calling	BISMARK, Bis-SNP, gemBS, MethylExtract	yes	yes			[15,45,50,52]
standard read alignment	bowtie2, BWA			yes		[53,54]
Normalization	DESeq2, MEDIPS, Diffbind			yes		[55,56,57]
Enrichment analysis	QSEA, RaMWAS, Diffbind			yes		[57,58,59]
Quality control	minfi, limma, wateRmelon				yes	[60,61,62]
Normalization	minfi, limma, wateRmelon				yes	[60,61,62]
Methylation calling (bvalues, mvalues)	minfi, wateRmelon				yes	[60,62]

**Table 2 cancers-14-00349-t002:** Methods for data imputation and exploratory analysis.

Process	Description	Method	Software	BulkBS-Seq	scBS-Seq	AE-Seq	BS-Arrays	Ref
Visualization	Variance decomposition	PCA	R	yes	yes	yes	yes	[68]
	Dimensionality reduction	MDS, t-SNE, NMF	MASS, stats, Rtsne, NMF	yes	yes	(yes)	(yes)	[69,70,71]
Clustering	Clustering (nearest neighbour)	k-means						
	Hierarchical clustering (un-/supervised)	hclust()	stats, cluster,	yes	yes	yes	yes	[72,73]
Imputation of missing data	Based on local spatial methylation correlation	Local likelihood smoothing	BSmooth	yes	(yes)			[74]
	Based on local spatial methylation correlations within and across cells and different genomic regions	glm, Bayesian clustering	Melissa		yes			[75]
	Based on local spatial methylation correlations within and across cells and different genomic regions	Bayesian clustering, hierarchical mixture model	Epiclonal	yes	yes			[76]
	Based on neighbouring CpG correlation and sequence composition	Deep neural network	DeepCpG		yes			[77]

**Table 3 cancers-14-00349-t003:** Methods for cell-type deconvolution and estimation of tumour purity.

Task	Class	Method	Software	Bulk BS-Seq	scBS-Seq	BS-Arrays	Ref
Remove unwanted variation (including batch effects)	Reference-free	Surrogate and independent surrogate variable analysis	SVA	yes		yes	[81]
		Remove unwanted variation	RUV, missMethyl			yes	[82,83]
Intra-sample cell type deconvolution	Reference-free, semi-reference-free	NMF using recursive QP	RefFreeEWAS	yes		yes	
	Reference based	Robust partial correlations, CIBERSORT, Houseman CP, COMBAT	HEpiDISH/EpiDISH			yes	[84]
		CIBERSORT	METHYLCIBERSORT	yes		yes	[85]
	Reference based using scRNAseq		EPISCORE	yes			
Estimate immune cell fraction in tumours	Reference based		MethylResolveR				[86]
Inference of tumour burden and tissue of origin from plasma cfDNA			CancerDetector	yes			
Estimate tumour purity from plasma cf-DNA	Reference-free	Concordance of neighbouring CpGs	CancerDetector				[87]
Estimate epipolymorphism, methylation entropy, clonal heterogeneity	Reference-free	Epiallele frequency	WSH	yes	(yes)		[88]

**Table 4 cancers-14-00349-t004:** Popular methods for differential DNA methylation.

Type	Method	Distribution	Software	Bulk BS-Seq	scBS-Seq	AE-Seq	BS- Arrays	Ref
DMC, DMR (predefined)	Fisher’s Exact test, logistic regression	Binomial (dispersion)	MethylKit	yes				[88]
DMC, DMR (predefined)	Likelihood ratio	Beta-binomial	MethylSig	yes				[90]
DMC, DMR (defines)	Wald test, linear regression	Beta-binomial (dispersion)	DSS	yes				[91]
DMC, DMR (defines)	local linear regression, smoothing, *t*-test similar	Binomial	BSseq (BSmooth),	yes				[74]
DMC, DMR (predefind)	Linear regression, *t*-test	Linear	RnBeads	yes	yes		yes	[92]
DMC, DMR (predefind)	glm, likelihood ratio	Negative-binomial (dispersion)	EdgeR	yes		yes		[93]
DMC, DMR (predefind)	glm, Wald test	Negative-binomial (dispersion)	DEseq2 (Diffbind)	yes		yes		[54]
DMC	non-parametric test, beta-regression	Gauss	limma				yes	[60]
DMC, DMR (defines)	local linear models, smoothing	Gauss	minfi (bump hunter, DMPfinder)				yes	[59]
DMC, DMR (defines)	local linear models, smoothing		DMRcate				yes	[94]
DMC, DMR (defines)	Linear models, combining subregions	Gauss	dmrff				yes	[95]

**Table 5 cancers-14-00349-t005:** Popular approaches for methylome segmentation.

Type	Method	Model	Software	Bulk DNA BS-Seq	Ref
UMR, LMR, HMR	Segmentation	3-State HMM	MethylSeekR	yes	[103]
PMD	Segmentation	3-State HMM	MethylSeekR	yes	
Hypo/Hypermethylated regions, DMR, PMR, PMD, AMR	Segmentation	2-State HMM, genomic windows	methPipe	yes	[100]

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
