# Peer review of "Experimental and Bioinformatic Approaches to Studying DNA Methylation in Cancer"

_cancers, 2022, doi:10.3390/cancers14020349_

Round 1

Reviewer 1 Report

This is a thoughtful and clearly presented review summarizing the implications of new forms of DNAm data collection and the bioinformatic methods that have to date been developed to analyse them. As data collection options in this field are expanding, this review provides a timely overview of the state of the science that is likely to be helpful to those new to the field of epigenetic cancer epidemiology.

However, the manuscript should be updated to address the following comments:

  • The authors sporadically mention tumour derived cell-free DNA methylation (e.g. page 8, line 238) which has been of growing interest for early cancer detection and tissue of origin analysis, but do not appropriately contextualise this data source in terms of the wider DNAm-cancer background framework or in terms of the experimental assays required for its detection.

  • Another area of increasing interest in DNAm data collection has been the emerging value of long-read sequencing (e.g. nanopore sequencing), approaches for which have not been discussed here. Some such approaches (e.g. Sakamoto et al 2021, doi: 10.1093/nar/gkab397) appear to fall beyond the bisulfite/MRSE/enrichment paradigm that the authors note.

  • Little to no mention is made of other DNA modifications such as 5-hydroxymethylcytosine which is detectable on some of the platforms the authors discuss. At a minimum, the impact such modifications have on DNAm detection should be discussed.

  • Multi-omic methods are mentioned in section 2 (DNA methylation assays) in terms of data capture, but methodological approaches for performing bioinformatic analysis of these data are not provided later. It would be appropriate such methods to be discussed. Also, the only ‘multi-omic’ data discussed by the authors is gene expression data. It would be reasonable to discuss chromatin accessibility (ATAC-seq), confirmation (hi-c), splicing, and/or proteomic data as well.

  • I assume that tables 1 and 2 are meant to summarize popular approaches being used in the field rather than providing anything approaching an exhaustive description. However, this framing should be made more clearly in the manuscript and ideally in the table titles themselves. Further, some important omissions have been made – perhaps most notably the DMR calling algorithm dmrff (https://www.biorxiv.org/content/10.1101/508556v1).

  • The sentence beginning on page 4, line 130 makes an unsupported claim of causality. However, more importantly the conclusion noted there doesn’t appear to follow from the figure panel cited.

  • Please clarify that “approximately Guassian” discussed on page 9, line 253 refers to the mean difference in DNAm levels between two categories. DNAm beta values are not normally distributed themselves, but a mean difference in beta values will be normally distributed as the sample size approaches the central limit theorem.

  • It would be valuable for the authors to comment on the validity of the approach described on page 9, line 256 of setting an absolute DNAm difference threshold between comparison groups for identification of differentially methylated positions.

  • Section 4.2 on cell heterogeneity should more clearly identify that reference-based deconvolution methods rely heavily on selection of an appropriate reference dataset and that failure to do so can bias results.

Minor comments:

  • Please ensure that the order of figures and figure panels mentioned in the text corresponds to the order they appear in the manuscript  (e.g. figure 1B mentioned after C/D).

  • Sentence on page 10, line 309 does not make grammatic sense

Author Response

We thank the reviewer for his/her positive response and his/her constructive and insightful comments.

Please see the attachment for a point-by-point respose.

Reviewer 2 Report

This paper describes 1) classical approaches to study DNA in whole cell populations and single cell and multi-Omics approaches, 2) focus on data analysis to identify methylation signatures, and 3) discuss challenges and quality of data.

The paper is well written and easy to understand.  I have only minor comments.

1. I found an inconsistent description of acronyms. For example, snmCseq and t-SNE are not described in the text.  It would be good to describe all acronyms used through the text for readers unfamiliar with the topic or the latest technologies.

2. In line 119-120 something is missing in the last sentence. Do the authors meant to write genomic copy number variations?

3. Paragraph 130-134 could be moved earlier in the text before the “2.1. Single cell and multi-omics approaches” section.

4. Tables 1, 2, 3, 4.1, and 4.2 have a column named Ref, but the reference is missing in that column. Was this a format issue?

Author Response

(The authors gave the same response as above.)
